# Clinical and Technical Validation of Novel Bite Force Measuring Device for Functional Analysis after Mandibular Reconstruction

**DOI:** 10.3390/diagnostics13040586

**Published:** 2023-02-05

**Authors:** Claudius Steffen, Katharina Duda, Dag Wulsten, Jan O. Voss, Steffen Koerdt, Susanne Nahles, Max Heiland, Sara Checa, Carsten Rendenbach

**Affiliations:** 1Department of Oral and Maxillofacial Surgery, Charité–Universitätsmedizin Berlin, Corporate Member of Freie Universität Berlin and Humboldt-Universität zu Berlin, Augustenburger Platz 1, 13353 Berlin, Germany; 2Julius Wolff Institute, Berlin Institute of Health, Charité–Universitätsmedizin Berlin, Augustenburger Platz 1, 13353 Berlin, Germany; 3BIH Charité Clinician Scientist Program, Berlin Institute of Health, Charité–Universitätsmedizin Berlin, BIH Biomedical Innovation Academy, Charitéplatz 1, 10117 Berlin, Germany

**Keywords:** mandibular reconstruction, bite force, functional analysis, sensors, surgical flaps, capacitive sensor

## Abstract

Bite force measuring devices that are generally suitable for edentulous patients or patients undergoing mandibular reconstruction are missing. This study assesses the validity of a new bite force measuring device (prototype of loadpad^®^, novel GmbH) and evaluates its feasibility in patients after segmental mandibular resection. Accuracy and reproducibility were analyzed with two different protocols using a universal testing machine (Z010 AllroundLine, Zwick/Roell, Ulm, Germany). Four groups were tested to evaluate the impact of silicone layers around the sensor: no silicone (“pure”), 2.0 mm soft silicone (“2-soft”), 7.0 mm soft silicone (“7-soft”) and 2.0 mm hard silicone (“2-hard”). Thereafter, the device was tested in 10 patients prospectively who underwent mandibular reconstruction using a fibula free flap. Average relative deviations of the measured force in relation to the applied load reached 0.77% (“7-soft”) to 5.28% (“2-hard”). Repeated measurements in “2-soft” revealed a mean relative deviation of 2.5% until an applied load of 600 N. Maximum bite force decreased postoperatively by 51.8% to a maximum mean bite force of 131.5 N. The novel device guarantees a high accuracy and degree of reproducibility. Furthermore, it offers new opportunities to quantify perioperative oral function after reconstructive surgery of the mandible also in edentulous patients.

## 1. Introduction

Tooth losses, temporomandibular joint dysfunctions, mandibular fractures and oral malignoma may all cause limitations in patients’ oral function and quality of life [1,2]. The oral function and chewing ability impact bone healing after mandibular fractures and reconstructions with free flaps due to strains acting differently on intersegmental gaps [3,4,5]. Available osteosynthesis systems consider patients’ anatomy and guarantee high mechanical integrity; however, individualization does not yet involve the adaption of the osteosynthesis design to the expected postoperative loads, which is known to be essential for successful healing. The high rate of incomplete osseous unions (around 35–45%) in patients who undergo mandibular reconstructions with osseous free flaps might be a result of this disregard [6,7]. For the design of biomechanically optimized osteosynthesis systems, pre- and postoperative oral function must be determined.

Clinical assessments of the oral function usually include analyses of mouth opening, speech, chewing ability and determination of maximum bite force [8,9,10]. Bite force is highly correlated with masticatory performance [11] and is therefore a relevant and objective approach to evaluate the oral function.

Several bite force measuring devices have been developed in the past, with many of them being commercially available [12]. However, they all present some limitations. Strain gauge transducers present a high thickness, which interferes with normal occlusion [13]. Piezoresistive and pressure transducers lack accuracy and reliability, while piezoelectric transducers are believed to have insufficient sensitivity [12]. The dental prescale system uses a pressure-sensitive film, which has high advantages due to the lack of interference with occlusion [14]. However, the device is not available outside of Japan, and analytical equipment is needed for data analysis. The low-cost sensor developed by Fasier-Wooller et al. also presents limitations due to high thickness [15]. Further options do not work in edentulous patients or are still in an experimental stage [16,17,18].

In patients who underwent mandibular reconstructions, previous studies were either not able to measure bite force in edentulous patients [9,19,20] or devices were used that interfered considerably with the occlusion [9,21,22]. One study used color-changeable chewing gum; however, this methodology did not produce quantitative force values, which are needed to objectify biomechanics [10]. Suitable bite force measuring devices for critical patient groups are therefore missing.

The aim of this study was to validate a new bite force measuring sensor (loadpad^®^), which is suitable for edentulous patients and patients who undergo mandibular reconstruction with free bone flaps. Firstly, the present study tests the reliability and reproducibility of the novel sensor. Secondly, the feasibility of bite force measurement in complex intraoral situations is evaluated.

## 2. Materials and Methods

### 2.1. Bite Force Measuring Sensor

A prototype of a capacity-type pressure-mapping sensor (prototype of loadpad^®^, novel GmbH, Munich, Germany) was produced by the company in consultation with our group. For the optimal compromise of pressure range (50 N/cm^2^) and size, a sensor dimension of 25 mm × 40 mm × 2.5 mm was chosen. The sensor is flexible, and electronics are in a separate location on the other side of the sensor (Figure 1). Bluetooth^®^ wireless and real-time data transfer at a sampling rate of 100 Hz is possible.

### 2.2. Validation Measurements

A universal testing machine (Z010 AllroundLine, Zwick/Roell, Ulm, Germany) was used for validation measurements (Figure 2).

One-point testing was performed using two different protocols in order to simulate bite forces. Different protocols were used to analyze different load situations.

Firstly, a continuously increasing, stepped vertical load (0.01 mm/s, holding time 5 s every 50 N until 600 N) was applied to analyze the impact of increasing pressure without pressure release. Secondly, a cyclic increasing vertical load (load in steps of 50 N (0.01 mm/s), holding time 5 s at each step, afterward complete load release for 5 s, until a load of 600 N) was performed for each group in order to analyze impact of resilience (Figure 3). Time, load and vertical displacement were detected continuously with the testing machine, while simultaneously recordings using the bite force device were performed.

Since the intraoral bite force measurement requires the use of a thin silicone layer around the sensor in order to avoid demolition of the sensor and guarantee an even load distribution, the validation tests included different set-ups using different silicones and silicone thicknesses.

The following four groups were compared using both protocols:“Pure”: no silicone layer (Figure 4A);“2-soft”: 2.0 mm silicone layer (S1, A-silicone, bisico, Bielefeld, Germany) on both sides of the device with a final hardness of approximately 72 Shore A (Figure 4B);“7-soft”: 7.0 mm silicone layer (S1, A-silicone, bisico, Bielefeld, Germany) on both sides of the device with a final hardness of approximately 72 Shore A (Figure 4C);“2-hard”: 2.0 mm silicone layer (Regidur^®^ i, A-silicone, bisico, Bielefeld, Germany) on both sides of the device with a final hardness of approximately 90 Shore A (Figure 4D).

### 2.3. Study Design

This prospective cohort study received approval from the Ethics Committee of the medical faculty of Charité—Universitätsmedizin Berlin (EA2/138/18). Inclusion criteria were patients who underwent a mandibular resection and immediate reconstruction with a microvascular free bone flap. Patients receiving surgery between July 2022 and November 2022 were included in the assessment. Maximum bite force was clinically tested in ten patients who underwent mandibular reconstruction using three repetitions for each run. Two patients were edentulous. Exclusion criteria were patients with incompliance, limited mouth opening or preoperative pathologies of the mandibular continuity such as pathological fractures.

Pre- and postoperative bite force measurements were performed. The postoperative measurement was performed 30 days (±4 days) after surgery.

### 2.4. Clinical Methodology

The clinical set-up is shown in Figure 5. Prior to measurements, a silicone layer (S1, A-silicone, bisico, Bielefeld, Germany) was applied simultaneously onto the lower and upper dental arch. A flat metal plate (45 mm × 45 mm × 5 mm) was then inserted in between in order to guarantee flat surfaces. The patients were instructed to bite onto the plate, and the setting time of the silicone was awaited (approximately 2 min). The sensor was connected to an electronic device (iPhone 13, Apple Inc., Cupertino, CA, USA) via Bluetooth^®^. Using the “loadpad” application (novel GmbH, Munich, Germany), the sensor was reset to zero without applying any load. The metal plate was removed from the mouth and replaced with the sensor. The recording was started in the application on the iPhone, and the patients were instructed to clench the teeth with maximum force three times. After the measurement, recording was stopped and saved. Results were exported from the electronic device to Microsoft Excel (Microsoft Excel, v.16.6, Microsoft Corporation, Redmond, WA, USA) for analysis.

### 2.5. Statistical Analysis

The collected data were compiled in a database (Microsoft Excel, v.16.6, Microsoft Corporation, Redmond, WA, USA). Descriptive statistics were performed using Statistical Package for Social Sciences program (SPSS version 27.0.1.0, IBM Corporation, Armonk, NY, USA).

## 3. Results

### 3.1. Validation Measurements

Validation of the measuring device was performed for all four groups. As shown in Figure 6, all groups presented a similar accuracy.

As shown in Table 1, cyclic loading resulted in higher relative deviations of the measured force compared to linear loading. The highest average relative deviation was seen in group “2-hard” under cyclic loading (5.28% until 600 N). The smallest average relative deviation was seen in group “7-soft” under linear loading (0.77% until 600 N). For linear load application, the highest average deviation was seen in group “2-hard” (2.17% until 600 N). The smallest average deviation for cyclic load application was seen in group “7-soft” (1.61% until 600 N). In groups “pure”, “7-soft” and “2-soft” relative deviation increased with increasing loads under dynamic load application. Two months of storage of the silicone of group “2-soft” resulted in a similar accuracy to the initial results of the same group.

Reproducibility was tested using group “2-soft” under cyclic dynamic loading with five repetitions until a load of 600 N. This group and protocol were used as this silicone thickness and protocol are most similar to clinical use. As indicated in Figure 7, there was a high degree of reproducibility when tests were repeated. The mean relative deviation was 2.5%. The highest range of relative deviations was identified at 450 N (range 1.2–2.6%).

### 3.2. Clinical Application

Maximum mean preoperative bite force was 272.9 N (±196.1), and maximum mean postoperative bite force was 131.5 N (±85.7) indicating a reduction by 51.8% measured 2–4 weeks after surgery (Table 2). There was a high variation in bite force differences (preoperative minus postoperative) between patients (range +4 N to −558.5 N). Two patients did not show any bite force reduction (patients 1 and 6). There was no clear trend in which repetition of each run (1, 2 or 3) led to the highest bite force.

## 4. Discussion

The results of the present study show that the novel device provides high accuracy and reproducibility and is suitable for bite force measurements in edentulous patients and patients after mandibular reconstruction.

The intraoral bite force measurement requires the use of a thin silicone layer around the sensor in order to avoid demolition of the sensor and guarantee an even load distribution. The silicone layer is between 0 mm and up to 5 mm in some areas. The validation tests therefore included different set-ups using different silicones and silicone thicknesses. The results indicate a high degree of accuracy independent of the chosen set-up. The maximum relative deviation from the applied load varied from 0% to 8.92%, which are acceptable values considering that deviations are higher at higher loads. Layers of soft silicone (“2-soft” and “7-soft”) slightly improved the accuracy of the measurements compared to the group “pure”, presumably by allowing a more even load distribution to the sensor. Despite slight variations, neither the thickness of the silicone layer nor the type of silicone resulted in relevant differences in the measured force. This is indicated by the average relative deviation of the measured force in relation to the applied load which demonstrated a range from 0.77% (group “7-soft”, continuous loading) to 5.28% (group “2-hard”, cyclic loading) until 600 N. Furthermore, a high rate of reproducibility is given at all loads up to 600 N, which is especially relevant when multiple measurements at different time points are intended.

Due to slightly better results, the soft silicone was used for the clinical application of bite force measurements. The feasibility of intraoral bite force measurement was tested in ten patients who underwent mandibular reconstruction, two of which were edentulous. Pre- and postoperative bite force measurements demonstrated a general decrease in the bite force by 51.8% about four weeks after reconstructive surgery. Familiarization with the device slightly influenced results due to higher maximum bite forces in the second or third repetition in many patients. Bite force determination with this newly developed device is feasible and accurate also in complex anatomical situations.

The prototype presented here measures force by capacitive sensing. Mechanical loads are registered by the sensor by a conversion to changes in capacitance, which allows the calculation of forces. Capacitive sensors for bite force measurement have only been described by one other group before [23,24]: Iwasaki et al. presented a validation of a capacitive-type pressure-mapping sensor for bite force determination; however, the bite force measurement in this study was only performed using a dental model [23]. A later clinical study by this group using the device excluded patients with missing occlusal contacts from the study [24]. This limitation is now solved by our measuring device, which allows bite force measurements also in edentulous patients.

Similar to the device described by Iwasaki et al., our device has the benefits of flexibility and thin proportions. Thus, interference with occlusion is limited to a small degree, although the device is not quite as thin as the dental prescale system [14]. The small size of the entire device including the portability allows easy use without further equipment. Connection to a smartphone is possible, and measurements can be repeated immediately. Besides the possibility to measure in edentulous patients, these are major advantages in comparison to the dental prescale system, which is not available in Europe.

Specifically focusing on patients with complex anatomy and muscle action after mandibular reconstruction, previous sensors used in these patients were not as generally applicable as the device presented here. Sakuraba et al. measured bite force in 24 patients and demonstrated that bite force decreased as the number of segments increased, presumably due to more muscle detachment [9]. The study was performed with an occlusal force meter. Thus, measurements in edentulous patients were not possible, and the thickness of the device resulted in a locked mouth closure [9]. Measuring devices that interfere significantly with occlusion have been previously stated as a major limitation when aiming to measure bite force accurately [12]. Moreover, Curtis et al. measured bite force in 10 patients after mandibular reconstruction using a device that separated dentition by approximately 10 mm [21,22]. Furthermore, this device only indirectly determined bite force using three strain gauges [22]. An innovative approach was described by Mochizuki et al., who evaluated bite force in patients after scapula flap reconstructions using color-changeable chewing gum [10]. However, this methodology did not produce comparable force values but only qualitative data [10]. Another study by Linsen et al. examined 26 patients after mandibular reconstruction with a device of 6 mm thickness and showed that bite force was significantly lower in resected jaw areas than in healthy ones. The measurement was also exclusively performed in dentulous patients [19]. In a further study by Linsen et al., showing advantages concerning functional outcomes in patients with reconstructions, edentulous patients could not be measured [25]. A thinner device (3 mm) was used by Maurer et al. who analyzed 20 patients, but measurements could again only take place in dentulous patients or patients with prostheses [20].

The bite force measuring device introduced here is therefore the first device that allows determination of the bite force in all patients with sufficient mouth opening after mandibular reconstruction.

Survival in cancer patients receiving mandibular reconstruction is the superior aim [26]. Besides that, a successful postoperative function is another main objective. High rates of pseudarthrosis after mandibular reconstruction are a known problem, and a higher rate of osseous non-union at the anterior segmental gap between the fibula flap and the mandible has been described [6,27]. Delayed healing hinders dental implantation and thus functional rehabilitation due to the need for re-osteosynthesis or the impossibility of plate removal [28,29]. The bite force as a measure of oral function is presumed to be a relevant variable influencing the biomechanics after mandibular reconstruction and its complications. From the long bone, the influence of micromovements on bone healing is well known [30]. Moreover, after mandibular reconstruction, differences in intersegmental movements between different gap sites have been described as dependent on the loading type and stiffness of fixation systems [31]. The extent of micromovements is directly influenced by oral function and bite force in particular. Many patients after mandibular resection are nourished with a feeding tube during the first days in order to avoid wound healing disorders and salivary fistula. Dysphagia, toothlessness and reduced mouth opening often require several weeks of liquid or soft diet or even nutrition with a percutaneous endoscopic gastrostomy tube, shielding the reconstructed mandible from load stimulation.

This reduced function has been presented by previous studies that described a maximum bite force of just up to 250 N after mandibular reconstruction [20,25]. In healthy adults, the maximum bite force is stated between 300 and 600 N and decreases with a reduction in the number of teeth [32]. Our results of 10 patients indicate that bite force may reach values of up to 300 N after four weeks after mandibular reconstruction. However, longitudinal studies analyzing the oral function continuously in the postoperative course are missing. For this reason, there is little knowledge concerning the ideal stiffness of osteosynthesis plates for mandibular reconstruction in order to reduce previously described complications [31]. Our presented bite force measuring device will allow continuous determination of the oral function in these patients over the postoperative course. Thus, more profound knowledge concerning postoperative biomechanics may be gained. Transfer to other complex anatomical situations such as morphological abnormalities can be expected [33].

As a limitation, it needs to be stated that the device was built as a prototype and is not generally commercially available yet. Despite low thickness, remaining interferences with occlusion cannot be excluded, and comparisons with other devices with different thicknesses need to be performed in the future. Patients with preoperative damage of the mandible, e.g., pathological fractures, incompliance or limited mouth opening, cannot be tested functionally with this device. For these patients, methods such as functional assessments using magnetic resonance imaging would need to be developed further [17].

## 5. Conclusions

This newly developed capacitive-type bite force measuring device guarantees accurate determination of bite force up to 600 N. There is a high degree of reliability and reproducibility, and the portable design offers easy use. The set-up allows determination of the oral function in most patients receiving mandibular reconstruction and other edentulous patients. Consequent determination of bite force in these patients allows increasing basic knowledge about the development of oral function of different types of reconstruction over the postoperative period. Thus, future individualized reconstructive therapy becomes more realistic.

## Figures and Tables

**Figure 1 diagnostics-13-00586-f001:**
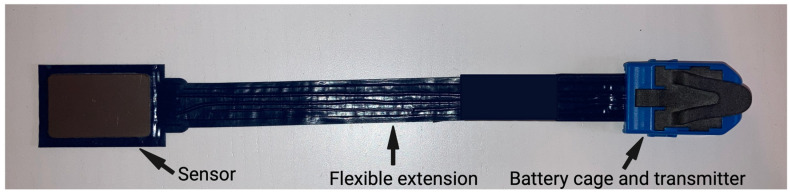
Bite force measuring sensor. Capacity-type pressure-mapping sensor including its flexible extension and battery cage. A transmitter allows wireless data transfer.

**Figure 2 diagnostics-13-00586-f002:**
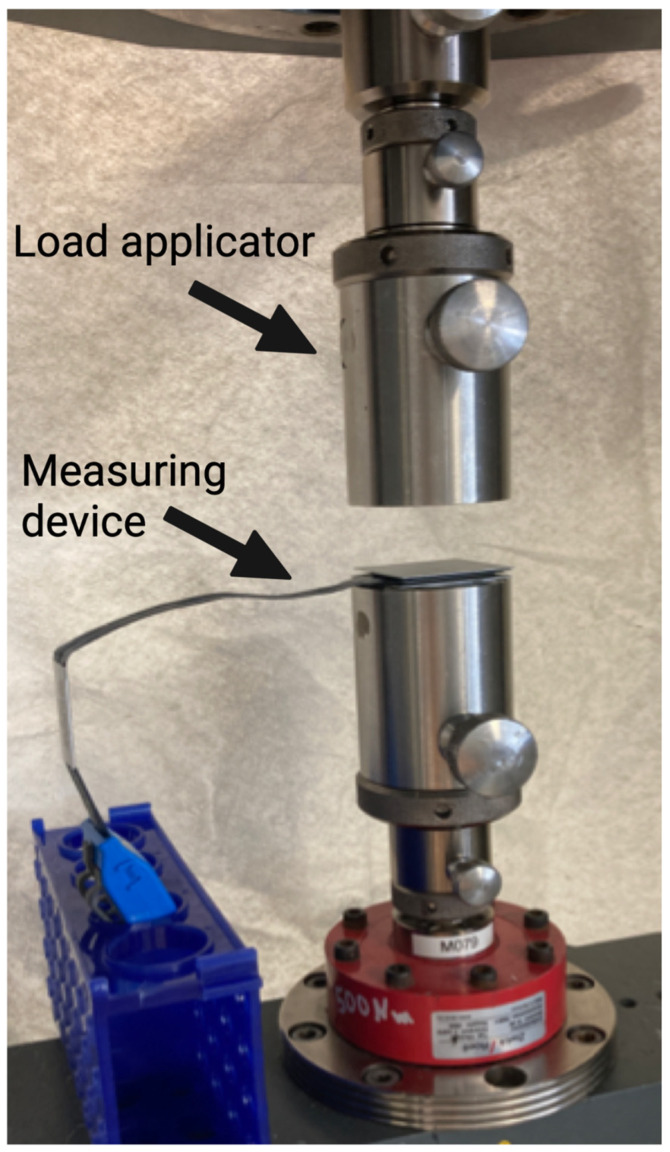
Set-up for validation measurement using the universal testing machine. Arrows indicate the load applicator and the measuring device. The measuring device is positioned without any preload and then loaded according to different protocols.

**Figure 3 diagnostics-13-00586-f003:**
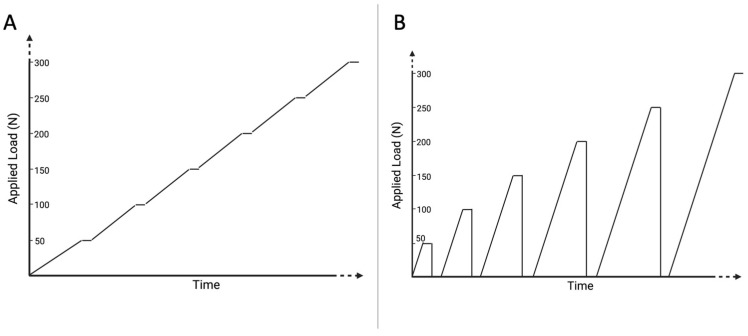
Schematic visualization of testing protocols. Simplified schematic demonstration of the two different protocols, which were used to simulate bite force measurements. A continuously increasing, stepped vertical load (**A**) and a cyclic increasing vertical load (**B**) were used for validation measurements. Created with Biorender.com.

**Figure 4 diagnostics-13-00586-f004:**
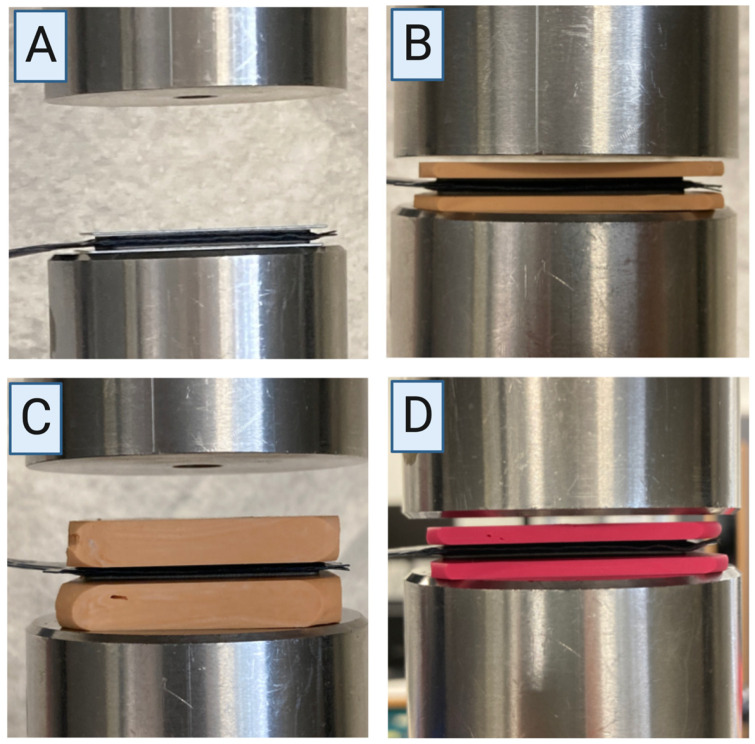
Demonstration of different set-ups. Four groups were analyzed: “Pure” (**A**), “2-soft” (**B**), “7-soft” (**C**), “2-hard” (**D**). Created using Biorender.com.

**Figure 5 diagnostics-13-00586-f005:**
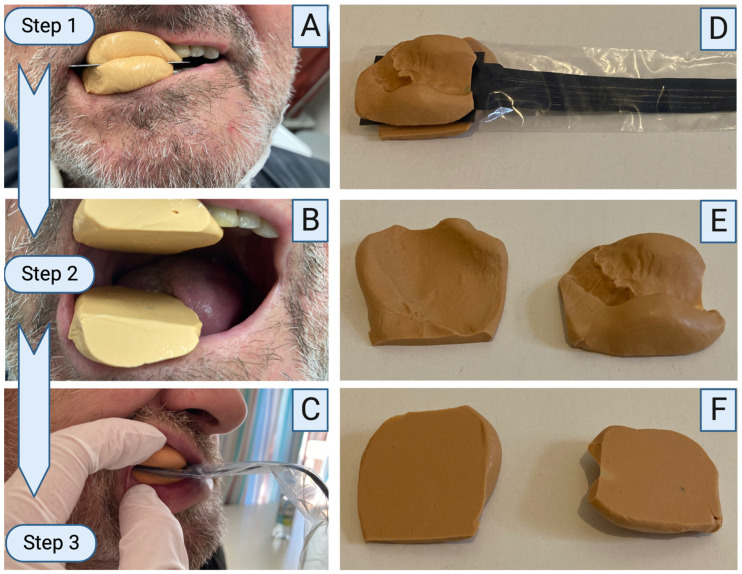
Bite force measuring device and its clinical application. Two silicone bulges are placed on the upper and lower jaw. The lower jaw is edentulous. A metal plate is placed in between, and the patient is asked to bite together. The patient is instructed to keep biting until the silicone hardened (**A**). The metal plate was removed, and two flat surfaces were created (**B**). The measuring device inside a protection cover is placed in between the flat silicone bulges, and the patient is instructed to clench the teeth together. Bite force can be recorded (**C**). After measurement, the complete set-up can be removed (**D**). The silicone allows complete individualization to the upper and lower jaw even in edentulous situations (**E**). Flat surfaces are directed to the sensor to guarantee protection of the sensor (**F**). Created using Biorender.com.

**Figure 6 diagnostics-13-00586-f006:**
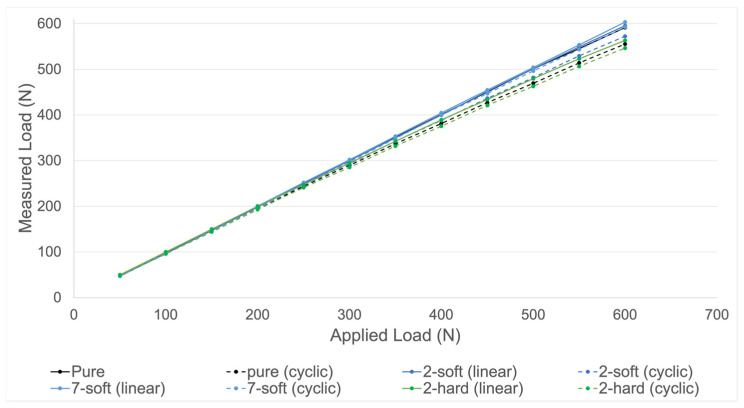
Validation measurements using four different set-ups. Measured load by the device in relation to the applied load by the testing machine is shown. Four different set-ups (pure, 2-soft, 7-soft, 2-hard) were tested with both linear (dotted lines) and cyclic (continuous lines) increasing loads (N) until 600 N.

**Figure 7 diagnostics-13-00586-f007:**
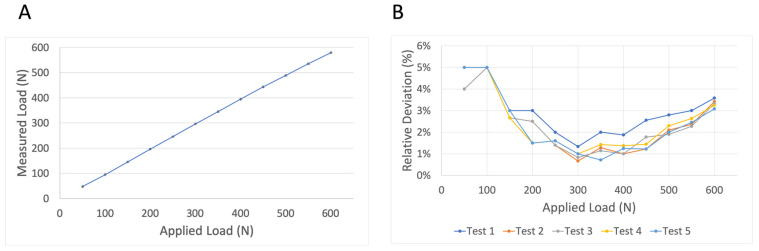
Reproducibility measurements. The average measured load of five repetitive measurements using group “2-soft” and cyclic dynamic loading until 600 N is presented (**A**). Standard deviations are presented by black error bars for each load step. The insignificant variances between repetitive measurements are further demonstrated by relative deviations (%) of the measured force in relation to the applied load by the testing machine (**B**).

**Table 1 diagnostics-13-00586-t001:** Validation measurements of different set-ups. Absolute forces measured by the device and relative deviations (%) of the measured force in relation to the applied load by the testing machine are presented. Four different set-ups (pure, 2-soft, 7-soft, 2-hard) were tested using both linear and dynamic increasing loads (N) until 600 N. Additionally, group 2-soft was re-tested after 2 months to identify the impact of storage.

Set Load (N)	Pure (Linear)	Pure (Cyclic)	2-Soft (Linear)	2-Soft (Cyclic)	7-Soft (Linear)	7-Soft (Cyclic)	2-Hard (Linear)	2-Hard (Cyclic)	2-Soft (Old) (Cyclic)
Measured load by force device (absolute values (N))									
50	47.5	49.0	47.0	49.0	49.0	48.0	50.0	49.0	47.0
100	97.0	97.0	97.0	97.5	100	95.5	100	96.5	94.0
150	147.5	145.0	148.0	146.0	149.5	146.5	150	144.0	144.0
200	198.5	194.5	198.5	196.5	200.5	195.5	200	193.0	194.5
250	249.5	244.0	249.0	247.0	252.0	247.5	247.5	241.0	243.5
300	300.5	290.0	300.0	293.5	302.0	297.5	295.0	285.5	294.5
350	351.0	337.0	350.0	342.5	353.5	348.5	342.5	332.0	343.0
400	401.0	381.5	400.0	388.5	404.5	399.5	389.5	375.5	393.0
450	450.5	427.0	451.5	436.5	454.5	447.0	434.0	421.0	441.5
500	501.0	469.5	501.0	482.0	504.0	496.5	479.5	463.0	486.5
550	545.5	514.0	549.0	529.5	553.5	543.5	523.0	506.5	533.0
600	591.5	555.5	595.5	572.5	603.5	592.5	563.5	546.5	575.5
Relative deviations from set load (%)									
50	5.00	2.00	6.00	2.00	2.00	4.00	0.00	2.00	6.00
100	3.00	3.00	3.00	2.50	0.00	4.50	0.00	3.50	6.00
150	1.67	3.33	1.33	2.67	0.33	2.33	0.00	4.00	4.00
200	0.75	2.75	0.75	1.75	0.25	2.25	0.00	3.50	2.75
250	0.20	2.40	0.40	1.20	0.80	1.00	1.00	3.60	2.60
300	0.17	3.33	0.00	2.17	0.67	0.83	1.67	4.83	1.83
350	0.29	3.71	0.00	2.14	1.00	0.43	2.14	5.14	2.00
400	0.25	4.63	0.00	2.88	1.13	0.12	2.63	6.13	1.75
450	0.11	5.11	0.33	3.00	1.00	0.67	3.56	6.44	1.89
500	0.20	6.10	0.20	3.60	0.80	0.70	4.10	7.40	2.70
550	0.82	6.55	0.18	3.73	0.64	1.18	4.91	7.91	3.09
600	1.42	7.42	0.75	4.58	0.58	1.25	6.08	8.92	4.08
Mean 600	1.16	4.19	1.08	2.68	0.77	1.61	2.17	5.28	3.22

**Table 2 diagnostics-13-00586-t002:** Bite force measurement. Results of bite force (N) measurements using the measuring device in patients who underwent mandibular reconstructions. Preoperative and postoperative (four weeks after surgery) measurements were performed. For each run (pre- and postoperative), three repetitions were performed.

Patient	Edentulous?	Bite Force (N)								
		Preoperative				Postoperative				Difference Postop._max_ − Preop._max_
		1	2	3	Maximum	1	2	3	Maximum	
1	No	39.5	45.0	43.5	45.0	44.5	37.0	40.0	44.5	−0.5
2	No	380.0	387.5	411.0	411.0	271.5	281.0	282.5	282.5	−128.5
3	No	292.0	335.5	315.0	335.5	162.5	164.5	180.0	180.0	−155.5
4	Yes	113.0	109.5	120.0	120.0	57.0	59.5	59.5	59.5	−60.5
5	Yes	40.0	46.5	48.5	48.5	38.5	38.0	41.0	38.5	−7.5
6	No	62.0	101.0	127.5	127.5	90.5	111.0	131.5	131.5	4.0
7	No	336.0	323.5	352.5	352.5	258.5	226.5	237.5	258.5	−94.0
8	No	173.0	249.0	248.5	249.0	60.0	60.5	71.5	71.5	−177.5
9	No	506.0	554.5	673.5	673.5	53.0	47.5	115.0	115.0	−558.5
10	No	285.0	339.0	366.5	366.5	43.5	104.0	130.5	130.5	−236.0

## Data Availability

Not applicable.

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
