# Peer review of "Clinical and Technical Validation of Novel Bite Force Measuring Device for Functional Analysis after Mandibular Reconstruction"

_diagnostics, 2023, doi:10.3390/diagnostics13040586_

Round 1

Reviewer 1 Report

Dear Authors,

Please find below some observations and recommendations concerning your article entitled” Validation of novel bite force measuring device for functional 2 analysis after mandibular reconstruction”.

Title and Authors

Please follow the MDPI authors' guidelines.

In the Abstract section:

- Please follow the MDPI authors' guidelines concerning the abstract structure (no more than 200 words should be included, without headings).

- Please rewrite the abstract section

In the Introduction section:

- Please indicate the reference number in square brackets

- Please include the null hypothesis instead of study’s hypothesis:” We hypothesized that the device presents a high reliability and allows determination of bite force in complex intraoral situations”. If there is no hypothesis to verify, please rewrite the phrase.

In the Materials and Methods section:

2.1. Bite force measuring sensor

- Figure 1 - in my opinion should be only the first image (fig.1.a)

2.2. Validation measurements

- line 96 – please replace Figure 1 with Figure 2 (previously it was Figure1b)

-line 97- please add the new Figure 2

- line 96 – please make a new paragraph” A one-point testing was performed...”

-line 104- please replace Figure 2 with Figure 3

-line 116- 123 please add in text Figure 4a-d (previously they were Figure1c-f)

-line 124- please add the new Figure 4

2.3. Clinical application

-line 131- please verify the correct number of edentulous patients to correspond with the results section (table 2- only two patients No.4 and 5, are edentulous)

- line 140-141- please give more details how you connect the sensor to the iPhone device (Bluetooth) and which application did you use.

2.4. Statistical analysis

-line 160- please replace SPSS withStatistical Package for Social Sciences program (SPSS version...)”

In the Results section:

- Table 2– please modify the structure to make it more understandable.

Eg.

Patient

Edentulous?

Bite force (N)

Preoperative

Postoperative

Difference

Postop.max- Preop.max

1

2

3

Maximum

1

2

3

Maximum

1

No

39.5

45.0

2

....10

In the Discussion section:

- If you decide to verify a null hypothesis, please provide some discussions.

In the References section: please follow the styles recommended for MDPI journals.

Author Response

Dear Reviewer,

thank you very much for your constructive comments. Please see the attachment for details.

Claudius Steffen

Reviewer 2 Report

In this paper, the authors presented a study entitled “ Validation of novel bite force measuring device for functional analysis after mandibular reconstruction.” with aim to validate a new bite force measuring sensor (loadpad)which is suitable for edentulous patients and patients who undergo mandibular reconstruction with free bone flaps. 

My recommendations are the following:

The abstract section is clear and well-performed. The introduction section is clear, on the other hand,  it is crucial to underline that metastasis involvement is the most important prognostic factors in terms of patient survival. Please consider : https://doi.org/10.3390/medicina58010054 . The aim of the study is clear.

The materials and methods section needs to be totally reorganized with subheadings: Please insert: Study design, sample size calculation (if needed) , Statistical analysis. Why were 10 patients  chosen to test this sensor? Has this prototype been patented? or will it be?

Discussion/Conclusions are clear and reflect results. 

  • In the first paragraph of the discussion, there is a repetition of the results. Please shorten or delete this paragraph. 
  • Moreover,The bibliography in the text is not formatted correctly. Check the journal guidelines

According to this Reviewer’s consideration, novelty and quality of the paper, publication of the present manuscript is recommended after major revision

Author Response

Dear Reviewer,

thank you very much for your comments. Please see the attachment for details.

Claudius Steffen

Round 2

Reviewer 1 Report

Dear authors,

Thank you very much for revising the manuscript according to my comments.

Reviewer 2 Report

The authors have adequately addressed the concerns of this reviewer.